# CAPSULE GRAPH NEURAL NETWORK

**Zhang Xinyi, Lihui Chen**
School of Electrical and Electronic Engineering
Nanyang Technological University, Singapore
`xinyi001@e.ntu.edu.sg, elhchen@ntu.edu.sg`

## ABSTRACT

The high-quality node embeddings learned from the Graph Neural Networks (GNNs) have been applied to a wide range of node-based applications and some of them have achieved state-of-the-art (SOTA) performance. However, when applying node embeddings learned from GNNs to generate graph embeddings, the scalar node representations may not suffice to preserve the node/graph properties efficiently, resulting in sub-optimal graph embeddings.

Inspired by the Capsule Neural Network (CapsNet) (Sabour et al., 2017), we propose the Capsule Graph Neural Network (CapsGNN), which adopts the concept of capsules to address the weakness in existing GNN-based graph embeddings algorithms. By extracting node features in the form of capsules, routing mechanism can be utilized to capture important information at the graph level. As a result, our model generates multiple embeddings for each graph to capture graph properties from different aspects. The attention module incorporated in CapsGNN is used to tackle graphs with various sizes which also enables the model to focus on critical parts of the graphs.

Our extensive evaluations with 10 graph-structured datasets demonstrate that CapsGNN has a powerful mechanism that operates to capture macroscopic properties of the whole graph by data-driven. It outperforms other SOTA techniques on several graph classification tasks, by virtue of the new instrument.

## 1 INTRODUCTION

GNN is a general type of deep-learning architectures that can be directly applied to structured data. These architectures are mainly generalized from other well-established deep-learning models like CNN (Krizhevsky et al., 2012) and RNN (Mikolov et al., 2010). In this paper, we mainly focus on Convolution-based Graph Neural Networks which attract increasing interest recently. Convolution operation can be embedded into Graph Neural Networks from spectral or spatial perspective. Bruna et al. (2013) defines the convolution operation in the Fourier domain which needs to calculate the eigendecomposition of the graph Laplacian. This method is computationally expensive and the filters they defined are non-spatially localized. Later, Henaff et al. (2015) introduces Chebyshev expansion of the graph Laplacian to avoid computing eigenvectors and Kipf & Welling (2017) proposes to do convolution within 1-step neighbor nodes to reduce the complexity. From the spatial perspective, Hamilton et al. (2017) and Zhang et al. (2018) propose to define a node receptive-field and do convolution within this field during which the information of each node as well as their neighbor nodes is gathered and new representation of each node is generated through an activation function. Both of these two perspectives perform well in node representation learning and a number of variants (Velikovi et al., 2018) are developed based on the convolution idea and some of them have proven to achieve SOTA in various tasks.

The success of GNN in node representation learning has inspired many deep-learning-based approaches to leverage on node embeddings extracted from GNN to generate graph embeddings for graph-based applications. However, during this procedure, the learned representation of each node will be considered as multiple individual scalar features instead of one vector. For example, Zhang et al. (2018) applies element-wise max-pooling to nodes embeddings when generating graph embeddings, Verma & Zhang (2018) generates graph embeddings by computing the element-wise covariance of all nodes. These operations indicate that the authors capture node features in the form

of scalar when they generate graph embeddings which may not suffice to preserve the node/graph properties efficiently.

To build high-quality graph embeddings, it is important to not only detect the presence of different structures around each node but also preserve their detailed properties such as position, direction, connection, etc. However, encoding these properties information in the form of scalar means activating elements in a vector one-by-one which is exponentially less efficient than encoding them with distributed representations. This has been identified discussed in Sabour et al. (2017). Inspired by CapsNet, we propose to extend scalar to vector during the procedure of applying GNN to graph representation learning. Compared with scalar-based neural network, vector-based neural network preserves the information of node/graph properties more efficiently. The technique for extracting features in the form of vectors is proposed in Hinton et al. (2011) and improved in Sabour et al. (2017) and Hinton et al. (2018). This technique is mainly devised for image processing. In their work, the extracted vector is referred to as capsule (a group of neurons in neural network), so we follow the same notation in our work. Introducing capsules allows us to use routing mechanism to generate high-level features which we believe is a more efficient way for features encoding. Compared with max-pooling in CNN in which all information will be dropped except for the most active one, routing preserves all the information from low-level capsules and routes them to the closest high-level capsules. Besides, this allows to model each graph with multiple embeddings and each embedding reflects different properties of the graph. This is more representative than only one embedding used in other scalar-based approaches.

In this paper, we propose Capsule Graph Neural Network (CapsGNN), a novel deep learning architecture, which is inspired by CapsNet and uses node features extracted from GNN to generate high-quality graph embeddings. In this architecture, each graph is represented as multiple embeddings and each embedding reflects the graph properties from different aspects. More specifically, basic node features are extracted in the form of capsules through GNN and routing mechanism is applied to generate high-level graph capsules as well as class capsules. In the procedure of generating graph capsules, an Attention Module can be applied to tackle graphs in various sizes. It also assigns different weights to each capsule of each node so that this model focuses on critical parts of the graph. We validate the performance of generated graph embeddings on classification task over 5 biological datasets and 5 social datasets. CapsGNN achieves SOTA performance on 6 out of 10 benchmark datasets and comparable results on the rest. T-SNE (Maaten & Hinton, 2008) is used to visualize the learned graph embeddings and the results show that different graph capsules indeed capture different information of the graphs.

## 2 BACKGROUND

Here, we provide a brief introduction to Graph Convolutional Networks (GCNs) (Kipf & Welling, 2017), routing mechanism in CapsNet and Attention mechanism which is used in CapsGNN.

### 2.1 GRAPH

By definition, a weighted directed graph can be represented by $\mathcal{G} = (\mathbb{V}, \boldsymbol{X}, \boldsymbol{A})$ where $\mathbb{V} = \{v_1, v_2, ...v_N\}$ is the set of nodes and $\boldsymbol{A} \in \{0, 1\}^{N \times N}$ is the adjacency matrix. If there is an edge from $v_i$ to $v_j$, then $A_{ij} = 1$ otherwise $A_{ij} = 0$. $\boldsymbol{X} \in \mathbb{R}^{N \times d}$ represents the features of each node. $d$ is the number of feature channels and $N$ is the number of nodes.

### 2.2 GRAPH CONVOLULTIONAL NETWORK

GCN, a widely used GNN architecture, is chosen as one of the key building blocks in our work. At each layer of the GCN, the convolution operation is applied to each node as well as its neighbors and the new representation of each node is computed through an activation function. This procedure can be written as:

$$\boldsymbol{Z}^{l+1} = f(\boldsymbol{T}\boldsymbol{Z}^l\boldsymbol{W}^l) \tag{1}$$

where $\boldsymbol{Z}^l \in \mathbb{R}^{N \times d}$ represents nodes features at the layer $l$, $d$ represents the number of feature channels and $\boldsymbol{Z}^0 = \boldsymbol{X}$, $\boldsymbol{W}^l \in \mathbb{R}^{d \times d'}$ is a trainable weights matrix which serves as a channel filter,

$f$ is a nonlinear activation function, $\boldsymbol{T} \in \mathbb{R}^{N \times N}$ is the information transform matrix and it is usually calculated from the adjacency matrix $\boldsymbol{A}$ for guiding the information flowing between nodes.

A complete GNN usually stacks $L$ layers to generate final nodes embeddings $\boldsymbol{Z}^L$. In the architecture proposed by Kipf & Welling (2017), at the $l$th layer of GCN, the extracted features of each node actually take all its adjacent nodes within $l$ steps into consideration. So $l$ can be considered as the size of the node receptive-field at this layer. This special property inspired us to use nodes features extracted from different layers to generate the graph capsules.

## 2.3 Capsule Neural Network

The concept of capsules is invented by Hinton's team (Hinton et al., 2011) and used recently in Sabour et al. (2017) and Hinton et al. (2018). CapsNet is designed for image features extraction and it is developed based on CNN. However, unlike traditional CNN in which the presence of feature is represented with scalar value in feature maps, the features in CapsNet are represented with capsules (vectors). In Sabour et al. (2017), the direction of capsules reflects the detailed properties of the features and the length of capsules reflects the probability of the presence of different features. The transmission of information between layers follows Dynamic Routing mechanism. The specific procedure of Dynamic Routing can be found in Appendix A for the completeness.

Inspired by CapsNet, the capsule mechanism is adopted and fused with GNN in our proposed Caps-GNN to generate graph capsules and class capsules on the basis of node capsules which are extracted from GNN. Dynamic Routing is applied to update weights between capsules from one layer to the next layer so that the properties captured by node capsules can be propagated to suitable graph capsules. Thus, each graph is modeled as multiple graph capsules, and then modeled as multiple class capsules. Different graph capsules reflect the properties of the graph from different aspects.

## 2.4 Attention Mechanism

Attention mechanism is widely applied in image (Zheng et al., 2017) and natural language processing domain (Gehring et al., 2016) where it is used to find the relevant parts of the input data to the task target. The main procedure of Attention mechanism is: 1) defining an attention measure which is used to measure the relevance of each part of the input data to the task target. 2) normalizing the generated attention value. 3) scaling each part with the normalized attention value.

In CapsGNN, we apply Attention mechanism for two purposes: 1) scaling each node capsule so that the graph capsules that are generated from different graphs are still comparable even though these graphs are vastly different in sizes. 2) guiding the model to focus on more relevant parts of graphs.

## 3 Capsule Graph Neural Network

In this section, we outline CapsGNN and show how it is used to generate high-quality graph capsules which then can be applied to graph classification task. Figure 1 shows a simplified version of CapsGNN. It consists of three key blocks:

1) Basic node capsules extraction block: GNN is applied to extract local vertices features with different receptive-field and then primary node capsules are built in this block. 2) High level graph capsules extraction block: Attention Module and Dynamic Routing are fused to generate multiple capsules for graphs. 3) Graph classification block: Dynamic Routing is applied again to generate class capsules for graph classification. The details of each block is explained in the following.

## 3.1 Basic Node Capsules

Firstly, the basic node features are extracted with GNN. Node degrees can be used as node attributes if nodes do not have attributes. We use the architecture improved by Kipf & Welling (2017) (GCN) as the node features extractor. The difference is that we extract multi-scale node features from different layers and the extracted features are represented in the form of capsules. The procedure can be written as:

$$\boldsymbol{Z}_j^{l+1} = f(\sum_i \tilde{\boldsymbol{D}}^{-\frac{1}{2}} \tilde{\boldsymbol{A}} \tilde{\boldsymbol{D}}^{-\frac{1}{2}} \boldsymbol{Z}_i^l \boldsymbol{W}_{ij}^l) \tag{2}$$

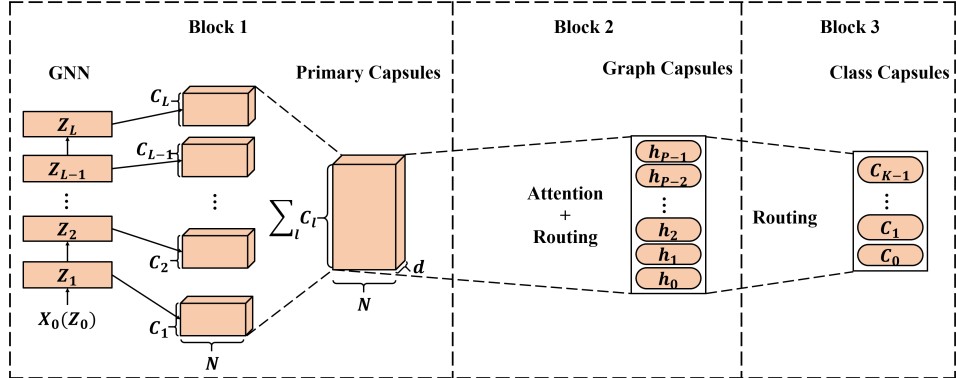

Figure 1: Framework of CapsGNN. At first, GNN is used to extract node embeddings and form primary capsules. Attention module is used to scale node embeddings which is followed by Dynamic Routing to generate graph capsules. At the last stage, Dynamic Routing is applied again to perform graph classification.

where $\boldsymbol{W}_{ij}^l \in \mathbb{R}^{d \times d'}$ is the trainable weights matrix. It serves as the channel filters from the $i$th channel at the $l$th layer to the $j$th channel at the $(l+1)$th layer. Here, we choose $f(\cdot) = tanh(\cdot)$ as the activation function. $\boldsymbol{Z}^{l+1} \in \mathbb{R}^{N \times d'}$, $\boldsymbol{Z}^0 = \boldsymbol{X}$, $\tilde{\boldsymbol{A}} = \boldsymbol{A} + \boldsymbol{I}$ [1] and $\hat{\boldsymbol{D}} = \sum_j \tilde{\boldsymbol{A}}_{ij}$. To preserve features of sub-components with different sizes, we use nodes features extracted from all GNN layers to generate high-level capsules.

## 3.2 HIGH-LEVEL GRAPH CAPSULES

After getting local node capsules, global routing mechanism is applied to generate graph capsules. The input of this block contains $N$ sets of node capsules, each set is $\mathbb{S}^n = \{\boldsymbol{s}_{11}, .., \boldsymbol{s}_{1C_1}, ..., \boldsymbol{s}_{LC_L}\}$, $\boldsymbol{s}_{lc} \in \mathbb{R}^d$, where $C_l$ is the number of channels at the $l$th layer of GNN, $d$ is the dimension of each capsule. The output of this block is a set of graph capsules $\boldsymbol{H} \in \mathbb{R}^{P \times d'}$. Each of the capsules reflects the properties of the graph from different aspects. The length of these capsules reflects the probability of the presence of these properties and the angle reflects the details of the graph properties. Before generating graph capsules with node capsules, an Attention Module is introduced to scale node capsules.

**Attention Module**. In CapsGNN, primary capsules are extracted based on each node which means the number of primary capsules depends on the size of input graphs. In this case, if the routing mechanism is directly applied, the value of the generated high-level capsules will highly depend on the number of primary capsules (graph size) which is not the ideal case. Hence, an Attention Module is introduced to combat this issue.

The attention measure we choose is a two-layer fully connected neural network $F_{attn}(\cdot)$. The number of input units of $F_{attn}(\cdot)$ is $d \times C_{all}$ where $C_{all} = \sum_l C_l$ and the number of output units equals to $C_{all}$. We apply node-based normalization to generate attention value in each channel and then scale the original node capsules. The details of Attention Module is shown in Figure 2 and the procedure can be written as:

$$scaled(\boldsymbol{s}_{(n,i)}) = \frac{F_{attn}(\tilde{\boldsymbol{s}_n})_i}{\sum_n F_{attn}(\tilde{\boldsymbol{s}_n})_i} \boldsymbol{s}_{(n,i)} \qquad (3)$$

where $\tilde{\boldsymbol{s}_n} \in \mathbb{R}^{1 \times C_{all}d}$ is obtained by concatenating all capsules of the node $n$. $\boldsymbol{s}_{(n,i)} \in \mathbb{R}^{1 \times d}$ represents the $i$th capsule of the node $n$ and $F_{attn}(\tilde{\boldsymbol{s}_n}) \in \mathbb{R}^{1 \times C_{all}}$ is the generated attention value. In this way, the generated graph capsules can be independent to the size of graphs and the architecture will focus on more important parts of the input graph.

---

[1] $\boldsymbol{I}$ represents identity matrix.

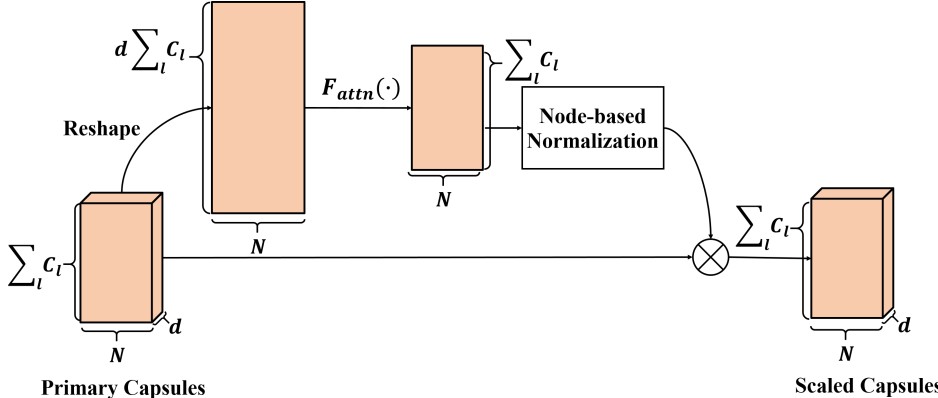

Figure 2: The structure of Attention Module. We first flatten primary capsules and apply two layer fully-connected neural network to generate attention value for each capsule. Node-based normalization (normalize each row here) is applied to generate final attention value. Scaled capsules are calculated by multiplying the normalized value with primary capsules.

After Attention Module, coordinate addition module can be used to preserve the position information of each node during the procedure of generating node capsule votes. Here, we introduce coordinate addition as an additional module and more details can be found in Appendix C.

The procedure of generating multiple graph capsules is summarized as follows:

1) Scale primary capsules: Apply Attention Module to scale primary capsules. The results of this module should be $S \in \mathbb{R}^{N \times C_{all} \times d}$.

2) Calculate votes: When calculating votes, capsules of different nodes from the same channel share the transform matrix. The result of this step is a set of votes $V \in \mathbb{R}^{N \times C_{all} \times P \times d}$ where $C_{all}$ denotes the number of channels. $P$ denotes the defined number of graph capsules.

3) Dynamic Routing Mechanism: High-level graph capsules are computed with the procedure introduced in Section 2.3 based on votes produced in previous steps.

## 3.3 CLASSIFICATION

This block is designed for graph classification using the graph capsules.

**Classification Loss**. Dynamic Routing is applied again over graph capsules to generate final class capsules $C \in \mathbb{R}^{K \times d}$, where $K$ is the number of graph classes. Here, we use margin loss function proposed in Sabour et al. (2017) to calculate the classification loss and it is computed as:

$$Loss_c = \sum_k \{T_k \max(0, m^+ - \|c_k\|)^2 + \lambda(1 - T_k)\max(0, \|c_k\| - m^-)^2\} \qquad (4)$$

where $m^+ = 0.9$, $m^- = 0.1$ and $T_k = 1$ iff the input graph belongs to class $k$. $\lambda$ is used to stop initial learning from reducing the length of all class capsules especially when $K$ is large.

**Reconstruction Loss**. Following Sabour et al. (2017), we use reconstruction loss as regularization method. Here, all class capsules are masked except the correct one and it is decoded with two fully-connected layer to reconstruct the input information. The information we reconstruct here is the histogram of input nodes. The procedure can be written as:

$$Loss_r = \frac{\sum_i MP_i(d_i - m_i)^2}{\sum_i MP_i} + \frac{\sum_i (1 - MP_i)(d_i - m_i)^2}{\sum_i (1 - MP_i)} \qquad (5)$$

where $m_i$ represents the number of nodes with the attribute $i$ appear in the input graph, $d_i$ is the corresponding decoded value. $MP_i = 1$ iff input graph contains nodes with attribute $i$. Equation 5 is used to prevent reducing reconstruction loss from setting all decoded value as 0 especially when most of the elements of the ground truth are 0.

The architecture details presented in section 3 describe the key design idea of CapsGNN which is based on the fusing of GNN and CapsNet. We also present a general comparison between CapsGNN with existing approaches in Appendix D.

# 4 EXPERIMENTS

We verify the performance of the graph embeddings extracted from CapsGNN against a number of SOTA approaches and some classical approaches on classification task with 10 benchmark datasets. Besides, we conduct experimental study to assess the impact of capsules in efficiency of encoding features of graphs. We also conduct brief analysis on the generated graph/class capsules. The experimental results and analysis is shown in the following. In addition to the analysis of the whole framework, we also provide a comparison experiment to evaluate the contribution of each module of CapsGNN with classification task. More details can be found in Appendix F.

## 4.1 GRAPH CLASSIFICATION

The goal of graph classification is to predict the classes these graphs belong to by analyzing the structure and nodes labels information of graphs. More specifically, given a set of labeled graphs $\mathbb{D} = \{(\mathcal{G}_1, y_1), (\mathcal{G}_2, y_2), \dots\}$ where $y_i \in \mathbb{Y}$ is the label of each graph $\mathcal{G}_i$. The objective of graph classification is to find a mapping $f$ such that $f : \mathcal{G} \to \mathbb{Y}$.

### 4.1.1 BASELINES METHODS

We compare CapsGNN with both kernel-based and deep-learning-based algorithms. The details are given as follows:

**Kernel-based Methods**: Three kernel-based algorithms, namely the Weisfeiler-Lehman subtree kernel (WL) (Shervashidze et al., 2011), the graphlet count kernel(GK) (Shervashidze et al., 2009), and the Random Walk (RW) (Vishwanathan et al., 2010). Typically, kernel-based algorithms first decompose graphs into sub-components based on the kernel definition, then build graph embeddings in a feature-based manner. Lastly, some machine learning algorithms (i.e., SVM) are applied to perform graph classification.

**Deep-Learning-based Methods**: Three types of deep-learning-based algorithms are selected:

1) Graph2vec (Narayanan et al., 2017), Deep Graph Kernel (DGK)(Yanardag & Vishwanathan, 2015) and AWE (Ivanov & Burnaev, 2018). Graph2vec, DGK and AWE require extracting sub-structures in advance while Graph2vec and AWE learn the representations of graphs in the manner of Doc2vec (Le & Mikolov, 2014), DGK applies Word2vec (Mikolov et al., 2013) to learn the similarity between each pair of sub-structures which will be used to build the graph kernel. Then kernel-based machine learning methods (i.e., SVM) are applied to perform graph classification. These three algorithms as well as kernel-based methods are all sub-components based and they all require two stages to do graph classification. So although Graph2vec, DGK and AWE apply learning approaches to learn the embeddings, we still consider them and other kernel-based algorithms as the same type in our experiments and we mainly compare our proposed architecture with the other remained methods which are all end-to-end and totally data-driven architectures.

2) PATCHY-SAN (PSCN)(Niepert et al., 2016). This method first sorts all nodes, then defines a receptive-field size for each node. These receptive-field are then filled with sorted neighbor nodes. Lastly, 1-D CNN is applied to perform graph classification.

3) GCAPS-CNN (Verma & Zhang, 2018), Dynamic Edge CNN (ECC) (Simonovsky & Komodakis, 2017) and Deep Graph CNN (DGCNN) (Zhang et al., 2018). These methods are all GNN-based algorithms. GCAPS-CNN first extract FGSD (Verma & Zhang, 2017) features for nodes that do not have attributes and then generate capsules for each node with higher-order statistical moment value of its neighbor nodes. At the last layer, they calculate covariance between all nodes to generate graph embeddings. ECC extracts node features on the condition of edge labels in GNN and then apply multi-scale pyramid structure to coarsen the graph. It uses average pooling at the last layer to generate graph embeddings. DGCNN generates nodes embeddings through a multi-layer GNN

and combine features extracted from all layers. Then they order the nodes based on the embeddings extracted from the last layer which is followed by 1-D CNN.

### 4.1.2 EXPERIMENTAL SET-UP

Five biological graph datasets: MUTAG, ENZYMES, NCI1, PROTEINS, D&D and five social network datasets: COLLAB, IMDB-B, IMDB-M, RE-M5K, RE-M12K (Yanardag & Vishwanathan, 2015) are used for our experimental study. Details of these datasets can be found in Appendix B.

We applied 10-fold cross validation to evaluate the performance objectively. Each time we use 1 training fold as validation fold to adjust hyper-parameters, 8 training fold to train the architecture and the remained 1 testing fold to test the performance. We stop training when the performance on the validation fold reaches to the highest. Then we use the accuracy on the test fold as our test result. The final result is the average of these 10 test accuracy. By default, we use the results reported in the original work for baseline comparison. However, in cases where the results are not available, we use the best testing results reported in Verma & Zhang (2018), Zhang et al. (2018) and Ivanov & Burnaev (2018). More details about experimental setting can be found in Appendix E.

### 4.1.3 CLASSIFICATION RESULT

Table 1 lists the results of the experiments on biological datasets, Table 2 lists the results of the experiments on social datasets. For each dataset, we highlight the top 2 accuracy in bold. Compared with all the other algorithms, CapsGNN achieves top 2 on 6 out of 10 datasets and achieves comparable results on the other datasets. Compared with all the other end-to-end architectures, CapsGNN achieves top 1 on all the social datasets.

Table 1: Experiment Result of Biological Dataset

| Algorithm | MUTAG | NCI1 | PROTEINS | D&D | ENZYMES |
|---|---|---|---|---|---|
| WL | 82.05±0.36 | **82.19±0.18** | 74.68±0.49 | **79.78±0.36** | 52.22±1.26 |
| GK | 81.58±2.11 | 62.49±0.27 | 71.67±0.55 | 78.45±0.26 | 32.70±1.20 |
| RW | 79.17±2.07 | >3days | 74.22±0.42 | >3days | 24.16±1.64 |
| Graph2vec | 83.15±9.25 | 73.22±1.81 | 73.30±2.05 | - | - |
| AWE | **87.87±9.76** | - | - | 71.51±4.02 | 35.77±5.93 |
| DGK | 87.44±2.72 | 80.31±0.46 | 75.68±0.54 | 73.50±1.01 | 53.43±0.91 |
| PSCN | **88.95±4.37** | 76.34±1.68 | 75.00±2.51 | 76.27±2.64 | - |
| DGCNN | 85.83±1.66 | 74.44±0.47 | 75.54±0.94 | **79.37±0.94** | 51.00±7.29 |
| ECC | 76.11 | 76.82 | - | 72.54 | 45.67 |
| GCAPS-CNN | - | **82.72±2.38** | **76.40±4.17** | 77.62±4.99 | **61.83±5.39** |
| **CapsGNN** | 86.67±6.88 | 78.35±1.55 | **76.28±3.63** | 75.38±4.17 | **54.67±5.67** |

CapsGNN achieves the SOTA performance on social datasets. More specifically, we are able to improve the classification accuracy by a margin of 2.78% and 5.30% on RE-M5K and RE-M12K respectively. This demonstrates that learning features in the form of capsules and modeling a graph to multiple embeddings is beneficial to capture macroscopic properties of graphs which are more important in classifying social networks. These results also consistent with the property of CapsNet, as it focuses more on extracting important information from children capsules by voting. However, applying routing to the whole graph leads to preserve all the information at a graph level and this property is not suitable to give prominence to individual fine structures which might be more important to biological datasets analysis. This results in less robust of CapsGNN on biological datasets. Despite this, the performance of CapsGNN in graph classification task still demonstrates its capability of graph representation especially its high potential of large graph dataset analysis.

Table 2: Experiment Result of Social Dataset

| Algorithm | COLLAB | IMDB-B | IMDB-M | RE-M5K | RE-M12K |
|---|---|---|---|---|---|
| WL | **79.02±1.77** | **73.40±4.63** | 49.33±4.75 | 49.44±2.36 | 38.18±1.30 |
| GK | 72.84±0.28 | 65.87±0.98 | 43.89±0.38 | 41.01±0.17 | 31.82±0.08 |
| DGK | 73.09±0.25 | 66.96±0.56 | 44.55±0.52 | 41.27±0.18 | 32.22±0.10 |
| AWE | 73.93±1.94 | **74.45±5.83** | **51.54±3.61** | **50.46±1.91** | 39.20±2.09 |
| PSCN | 72.60±2.15 | 71.00±2.29 | 45.23±2.84 | 49.10±0.70 | **41.32±0.42** |
| DGCNN | 73.76±0.49 | 70.03±0.86 | 47.83±0.85 | 48.70±4.54 | - |
| GCAPS-CNN | 77.71±2.51 | 71.69±3.40 | 48.50±4.10 | 50.10±1.72 | - |
| **CapsGNN** | **79.62±0.91** | 73.10±4.83 | **50.27±2.65** | **52.88±1.48** | **46.62±1.90** |

## 4.2 Efficiency of Capsules

The main objective of this experiment is to examine the efficiency of capsules in encoding graph features. More efficient in feature encoding here means representing more information with the similar number of neurons. We construct a scalar-based neural network for each CapsGNN and then compare the CapsGNN with its related scalar-based architecture by comparing their training and testing accuracy on graph classification task to demonstrate the efficiency in feature representation. More specifically, these scalar-based architectures are designed by replacing the graph capsules block and the class capsules block in CapsGNN with fully-connected layers (FC). In this case, the only difference between each pair of CapsGNN and its corresponding scalar-based architecture is that CapsGNN represents features with vectors and uses routing to propagate information between layers while the scalar-based architecture encodes features with scalar values.

In this experiment, the number of layers of GNN is set as $L = 3$, the number of channels at each layer is all set as $C_l = 2$. We construct different CapsGNNs by adjusting the dimension of nodes ($d_n$) and graphs ($d_g$) capsules and the number of graph capsules ($P$). The size of FC in scalar-based architectures is adjusted based on the size of CapsGNNs so that they have comparable number of trainable weights. Other hyper-parameters are the same as Appendix E. The details of the tested architectures are shown in Table 3. Besides, NCI1 dataset, which has more than 4000 graphs, is used for the test. The accuracy of NCI1 on various architectures can be found in Figure 3.

Table 3: Details of Tested Architectures in Efficiency Evaluation Experiment

| | 2-4-2 | 2-4-4 | 2-4-8 | 2-4-16 | 4-4-2 | 4-4-4 | 4-4-8 | 4-4-16 |
|---|---|---|---|---|---|---|---|---|
| No. trainable Para.(Caps) | 208 | 367 | 688 | 1328 | 448 | 704 | 1216 | 2240 |
| No. trainable Para.(Scalar) | 216 | 370 | 692 | 1336 | 452 | 712 | 1232 | 2246 |
| Dim Node Feat.(Both) | 2 | 2 | 2 | 2 | 4 | 4 | 4 | 4 |
| Dim Graph emb.(Caps) | $2 \times 4$ | $4 \times 4$ | $8 \times 4$ | $16 \times 4$ | $2 \times 4$ | $4 \times 4$ | $8 \times 4$ | $16 \times 4$ |
| Dim FC.(Scalar) | 12 | 23 | 48 | 92 | 10 | 20 | 40 | 79 |

In Table 3 and Figure 3, the setting of different architectures is represented as $d_n$-$d_g$-$P$ . Here, we choose the simplest setting (*2-4-2*) as an example: *2-4-2* means that the dimension of nodes capsules is $d_n = 2$, the dimension of graph and class capsules is $d_g = 4$ and the number of graph capsules $P$ equals to 2. Besides, we set the dimension of FC of its corresponding scalar-based architecture as 12 so that they have comparable number of trainable weights. In this case, each graph is modeled as 2 4-dimensional graph embeddings in the CapsGNN or 1 12-dimensional graph embedding in its corresponding scalar-based architecture. Both architectures are sub-optimal to represent the whole dataset while CapsGNN can still reach higher accuracy compared with the scalar-based architecture.

As we can see from Figure 3, the test accuracy of Caps-based architectures (CapsGNN) is higher than the corresponding scalar-based architectures in all settings. For the training accuracy, when the

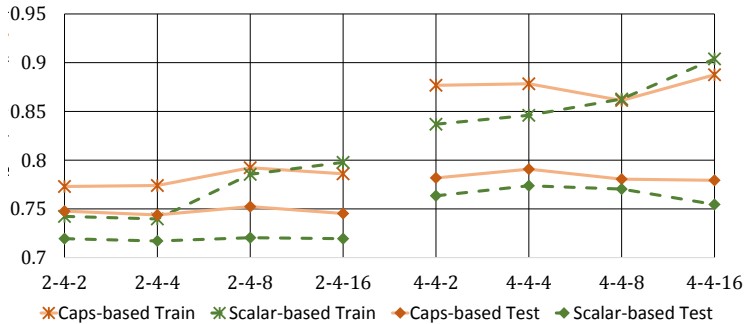

Figure 3: Comparison of efficiency in feature representation. The horizontal axis represents the setting of tested architectures. The vertical axis represents classification accuracy on NCI1.

dimension of FC is slightly higher than the dimension of graph embeddings in CapsGNN, CapsGNN can still reach higher accuracy which indicates that CapsGNN is more powerful in representing the whole dataset. When we keep increasing the number of graph capsules in CapsGNN and enlarging the dimension of FC in scalar-based architectures, the difference between the dimension of graph embeddings and the size of FC becomes larger, their training accuracy will be closer. It is noted that the training accuracy of scalar-based architectures is slightly higher than the CapsGNNs when the dimension of FC is about 20% larger than the dimension of graph capsules. In this experiment, we use extremely simple architectures on purpose to simulate the situation where we need to model complex datasets with relatively simple architectures. Since each pair of CapsGNN and its corresponding scalar-based architecture have similar structure and comparable number of trainable weights, the higher training accuracy and testing accuracy of CapsGNN demonstrate its efficiency in feature encoding and its strong capability of generalization.

## 4.3 GRAPH CAPSULES ANALYSIS

CapsGNN leverages on capsules idea to get multiple embeddings for each graph so that complex information underlying graphs can be captured more effectively. To explore the properties of the extracted graph/class capsules, we plot the graph distribution based on capsules extracted from different channels with t-SNE. Due to space constrain, we only take REDDIT-M12K as an example.

Table 4: Visualization of Graph Capsules

| Subreddit | Channel 1 | Channel 2 | Channel 11 | Channel 14 |
|---|---|---|---|---|
| atheism & IAmA | | | | |
| atheism & mildlyinteresting | | | | |

We choose to depict the distribution of graphs which are generated from 3 categories, namely *atheism* , *IAmA* and *mildlyinteresting* with capsules extracted from the 1st, 2nd, 11th, 14th channel of graph capsules. As we can see from Table 4, different channels of capsules represent different aspects of graph properties. *atheism* and *IAmA* can be discriminated obviously with capsules extracted from the 11th and the 14th channels while they are hard to be separated with capsules extracted

from the 1st and the 2nd channels. However, *atheism* and *mildlyinteresting* can be discriminated with the capsules extracted from the 1st and the 2nd channels while they are mixed in the 11th and the 14th channels which is opposite to the case of *atheism* and *IAmA*. This phenomenon can also be observed in other multi-class datasets. It is still hard to figure out the specific aspects these capsules focus on. However, compared with scalar-based neural networks, modeling an object with multiple embeddings makes it possible to explore the meaning of each channel which may lead the model to learn more interpretable embeddings in the future.

Table 5: Visualization of Class Capsules

| Subreddit | atheism | IAmA | mildlyinteresting | Concatenate |
|---|---|---|---|---|
| atheism(g) & IAmA(r) & mildlyinteresting(b) | | | | |

As we can see from Table 5, different class capsules focus on different classification-related graph properties. For example, the capsules that represent *atheism* (first column) can well discriminate *athesism* (red) from the other two types of graphs while *IAmA* (green) and *mildlyinteresting* (blue) are mixed in this channel. The similar phenomenon can also be found in other class capsules. Besides, when we concatenate the capsules of these three classes together, three types of graphs can be well discriminated with the concatenated capsules which also directly reflect the classification performance. This property is quite different from standard scalar-based architectures where each graph is modeled with only one graph embedding[2]. By introducing the concept of capsules, the graph and class capsules can not only preserve classification-related properties of each graph (reflected with the length of class capsules) but also other properties information (reflected with the angle of class capsules). The generated class capsules can also be useful in other follow-up work and we leave this to be explored in the future.

## 5 CONCLUSION

We have proposed CapsGNN, a novel framework that fuses capsules theory into GNN for more efficient graph representation learning. Inspired by CapsNet, the concepts of capsules are introduced in this architecture to extract features in the form of vectors on the basis of nodes features extracted from GNN. As a result, one graph is represented as multiple embeddings and each embedding captures different aspects of the graph properties. The generated graph and class capsules can preserve not only the classification-related information but also other information with respect to graph properties which might be useful in the follow-up work and we leave this to be explored in the future. We believe this is a novel, efficient and powerful data-driven method to represent high-dimensional data such as graphs. Our model has successfully achieved better or comparable performance when compared with other SOTA algorithms on 6 out of 10 graph classification tasks especially on social datasets. Compared with similar scalar-based architectures, CapsGNN is more efficient in encoding features and this would be very beneficial for processing large datasets.

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

## A  SPECIFIC PROCEDURE OF ROUTING

The specific procedure of routing is shown in Algorithm 1.

---

**Algorithm 1** Dynamic routing mechanism returns parent capsules $\boldsymbol{H}$ given children capsules $\boldsymbol{S}$, a set of trainable transform matrices $\mathbb{W}$ and the number of iterations $t$.

---

1: **procedure** DYNAMIC ROUTING$(t, \boldsymbol{S}, \mathbb{W})$
2:     for all children capsule $i$ : $\boldsymbol{v}_{j|i} = \boldsymbol{s}_i^T \boldsymbol{W}_{ij}$
3:     for all children capsule $i$ to all parent capsule $j$: $r_{ij} \leftarrow 0$
4:     **for** $t\ iterations$ **do**
5:         for all children capsule $i$: $\tilde{\boldsymbol{r}}_i \leftarrow softmax(\boldsymbol{r}_i)$
6:         for all parent capsule $j$: $\boldsymbol{h}_j \leftarrow \sum_i \tilde{r}_{ij} v_{ij}$
7:         for all parent capsule $j$: $\tilde{\boldsymbol{h}}_j \leftarrow squash(\boldsymbol{h}_j)$
8:         for all children capsule $i$ to all parent capsule $j$: $r_{ij} \leftarrow r_{ij} + \tilde{\boldsymbol{h}}_j^T \boldsymbol{v}_{ij}$
9:     **end for**
10:     **return** $\tilde{\boldsymbol{h}}_j$
11: **end procedure**

---

## B  DETAILS OF EXPERIMENTAL DATASETS

The details of benchmark datasets we use in our experiment is shown in Table 6.

Table 6: Dataset Description

| Dataset | Source | Graphs | Classes | Nodes Avg. | Edges Avg. | Nodes Labels |
|---------|--------|--------|---------|-----------|-----------|-------------|
| MUTAG | Bio | 188 | 2 | 17.93 | 19.79 | 7 |
| ENZYMES | Bio | 600 | 6 | 32.46 | 63.14 | 6 |
| NCI1 | Bio | 4110 | 2 | 29.87 | 32.30 | 23 |
| PROTEINS | Bio | 1113 | 2 | 39.06 | 72.81 | 4 |
| D& D | Bio | 1178 | 2 | 284.31 | 715.65 | 82 |
| COLLAB | Social | 5000 | 3 | 74.49 | 4914.99 | - |
| IMDB-B | Social | 1000 | 2 | 19.77 | 193.06 | - |
| IMDB-M | Social | 1500 | 3 | 13 | 131.87 | - |
| REDDIT-M5K | Social | 4999 | 5 | 508.5 | 1189.74 | - |
| REDDIT-M12K | Social | 11929 | 11 | 391.4 | 456.89 | - |

## C  COORDINATE ADDITION

After Attention Module, coordinate addition can be used to preserve the position information of each node during the procedure of generating node capsule votes. The details of Coordinate Addition module can be found in Figure 4. This module is not necessary in some datasets. Here, we propose this module as a selective optimization. When the GNN goes deeper, the extracted nodes features contain more specific position information of each node. Inspired by Zhang et al. (2018) where the node embeddings learned from the last layer of GNN are taken to order all nodes, we also take the capsules extracted from the last layer of GNN as the position indicators of corresponding nodes by concatenating it with each capsule of the node. The procedure of calculating votes with node position indicators can be written as:

$$\boldsymbol{v}_{(n,i)j} = [\boldsymbol{s}_{(n,i)}^T \boldsymbol{W}_{ij}^n \| \boldsymbol{s}_{(n,C_{all})}^T \boldsymbol{W}_j^p] \tag{6}$$

where $\boldsymbol{v}_{(n,i)j} \in R^{1 \times (d_n + d_p)}$ represents the node capsule vote from the $i$th channel of the $n$th node to the $j$th channel of graph capsules. $\boldsymbol{W}_{ij}^n \in R^{d \times d_n}$ and $\boldsymbol{W}_j^p \in R^{d \times d_p}$ are the transform matrices. $\boldsymbol{s}_{(n,i)}$ is the same as introduced in Section 3.2 and $\|$ represents concatenate operation.

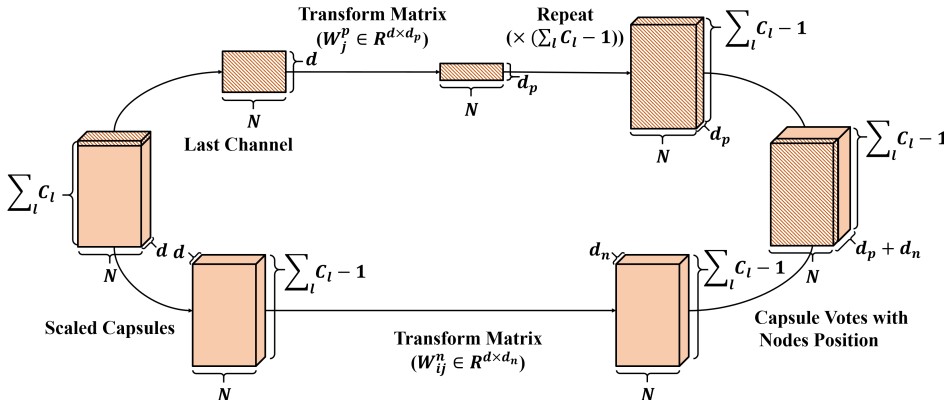

Figure 4: The structure of Coordinate Addition Module. We take the capsules extracted from the final layer of GNN as the position indicators of corresponding nodes by concatenating it with each capsule of the node. The node capsules votes generated in this way contain more position information.

## D    DIFFERENCES FROM EXISTING APPROACHES

Here, we present a general comparison between CapsGNN with existing approaches.

1) Compared with Atwood & Towsley (2016), Simonovsky & Komodakis (2017) and Zhang et al. (2018) (GNN-based graph representation learning architectures), CapsGNN represents node features in the form of capsules. This is helpful to preserve the properties information contained in nodes more efficiently when generating graph embeddings. Besides, each graph is modeled as multiple embeddings in CapsGNN instead of only one embedding used in other approaches. This allows us to capture information of graphs from different aspects. The second difference is that, in these approaches, each part of the graph is given equal importance. However, the attention mechanism used in CapsGNN allows it to assign various weights to different nodes. This leads the model to focus on critical parts of input graphs. Lastly, different from Atwood & Towsley (2016) and Simonovsky & Komodakis (2017), CapsGNN and Zhang et al. (2018) uses node features extracted from multiple layers of GNN so that different size of receptive-fields are applied to preserve more information.

2) GCAPS-CNN proposed by Verma & Zhang (2018) also introduced capsule-related concept into graph representation learning. However, they generate capsules in a feature-based manner instead of learning capsules as distributed embeddings. More specifically, when they extend a scalar feature to a capsule for the node $n$, $P$ higher-order statistical moment value is calculated based on its neighbor nodes and these $P$ value is concatenated to a $P$-dimensional capsule. Between layers, GCAPS-CNN performs dimension reduction to compress capsules back to scalar features, which defeats the purpose of having capsules in the first place. CapsGNN learns each dimension of capsules in a data-driven manner and apply routing mechanism between layers to preserve the learned meaning of each capsule. This also allows us to preserve multiple properties information contained in nodes more efficiently especially when generating graph embeddings.

3) Compared with CapsNet proposed by Sabour et al. (2017) which works well in image processing domain, CapsGNN needs to handle more complex situations when handling graphs. In image processing domain, the size of the input images can be standardized by resizing the images. However, it is not possible to simply resize the graphs. So, we introduced an additional Attention Module to tackle graphs that are vastly different in sizes and preserve important parts of graphs. We also propose to use features extracted from all layers of GNN since it is hard to define a suitable receptive-field size for graphs. Furthermore, compared with the architecture of CapsNet, CapsGNN has one additional graph capsules layer which is used to learn multiple graph embeddings and these embeddings reflect different aspects of graph properties which is valuable in future research. To the best of our knowledge, we are the first one to model a graph as multiple embeddings in the form of distributed capsules and we believe this approach of learning representations has a high potential

for other complex data analysis which is not limited to graphs. Besides, CapsGNN has different explanation of linear transformation. In CapsNet, by applying a linear trainable transformation to pose vectors, the spatial relationship between object parts and the whole object can be well modeled. However, by applying linear trainable transformation, CapsGNN is simply computing the prediction vectors from nodes-level representations to graph-level representations. This transform matrix is not trying to model the change of viewpoint or capture viewpoint invariant knowledge but to model the relationship between the properties of nodes and the properties of the whole graph.

# E  EXPERIMENTAL SETTING

The same architecture settings are used in CapsGNN for all datasets to show its robust performance. For the node capsules extraction, the GCN has 5 layers ($L = 5$), the number of channels at each layer is set as the same which is 2 ($C_l = 2$). The number of graph capsules is fixed as 16 ($P = 16$). The dimension of all capsules are set as 8 ($d = 8$). The number of units in the hidden layer of Attention Module is set as $\frac{1}{16}$ of the number of input units. The number of iterations in routing is set as 3. During training stage, we simultaneously reduce $Loss_c$ and $Loss_r$ and we scale $Loss_r$ with 0.1 so that the model focuses on classification task. $\lambda$ is set as 0.5 and 1.0 for multi-class classification and binary classification respectively. As for the node attributes in different datasets, considering that REDDIT-M5K and REDDIT-M12K are large-scale datasets with widely distributed nodes degree, we set the attributes of all nodes in these two datasets as the same which means we consider the initial node embeddings for all the nodes as the same to avoid over-fitting. For the remained relatively small datasets, both of the node degree and other node attributes are sent to CapsGNN as node features to speed up the training and we apply dropout($dropput\_rate$ is 0.3) to the input node features to improve the learning performance. The settings are summarized in the Table 7:

Table 7: Experimental Setting for Graph Classification

| Hyper-parameter description | Notation | Value |
|---|---|---|
| Number of GCN layers | $L$ | 5 |
| Number of channels at each layer | $C_l$ | 2 |
| Number of graph capsules | $P$ | 16 |
| Dimension of all capsules | $d$ | 8 |
| Number of iterations in routing | $t$ | 3 |
| Value used to scale $Loss_r$ | $r$ | 0.1 |
| Value used to balance the loss from the positive and negative output | $\lambda$ | 0.5 for multi-class classification
1.0 for binary classification |

# F  CONTRIBUTION OF EACH MODULE

In addition to evaluate the performance of the whole architecture on the classification task, we also provide detailed study to quantify the contributions made by each module in CapsGNN on the

classification task. The six comparison architectures set-up is shown as below and the settings of the hyper-parameters are the same as Section 4.1.2.

1) **CapsGNN** (GCN + Attention + Routing + Reconstruction): Basic CapsGNN.

2) **CapsGNN-Coord** (GCN + Attention + Coordinate + Routing + Reconstruction): Basic Caps-GNN with Coordinate Addition module.

3) **CapsGNN-Avg** (GCN + Average + Routing + Reconstruction): The Attention module in basic CapsGNN is replaced with the Average module.

4) **CapsGNN-noRout** (GCN + Attention + Reconstruction): In this architecture, we will fix the similarity coefficients between all the capsules from one layer to the next layer as the same so that each children capsule will be equally routed to all the parent capsules.

5) **CapsGNN-noRecon** (GCN + Attention + Routing): In this architecture, we directly remove the Reconstruction loss module.

6) **CapsGNN-Avg-noRout** (GCN + Routing): In this architecture, we replace the Attention module in basic CapsGNN with the Average module and fix the similarity coefficients between all the capsules from one layer to the next layer as the same.

Table 8: Validation Accuracy Comparison of Each Module

| Architecture | COLLAB | IMDB-B | PROTEINS | NCI1 | D&D |
|---|---|---|---|---|---|
| CapsGNN | 79.77±1.15 | 74.42±2.20 | **77.27±2.58** | **79.23±1.88** | 76.16±4.19 |
| CapsGNN -Coord | 80.00±1.22 | **74.81±2.96** | 76.58±2.42 | 79.04±1.93 | **77.41±3.33** |
| CapsGNN -Avg | **79.61±0.81** | **73.29±2.66** | 76.54±2.98 | 78.36±1.95 | 74.44±3.46 |
| CapsGNN -noRout | **80.48±0.86** | 74.11±2.94 | 77.18±2.94 | 77.62±1.15 | 75.28±4.17 |
| CapsGNN -noRecon | 80.44±0.88 | 74.03±2.11 | 76.69±1.78 | 78.23±2.41 | 74.70±3.12 |
| CapsGNN -Avg-noRout | 80.15±1.02 | 74.23±3.47 | **75.94±2.61** | **76.25±2.40** | **73.93±3.56** |

The validation accuracy of each architecture is shown in Table 8 where we highlight the highest and lowest accuracy respectively. As we can see from the Table, IMDB-B and D&D reach better performance with CapsGNN-Coord which indicates the effectiveness of Coordinate Addition Module. However, the performance of social datasets is still comparable across all types of architectures. On the other hand, the performance of biological datasets(NCI1, PROTEINS, D&D) is more sensitive to the introduced modules in each architecture. The highest accuracy of NCI1, PROTEINS is achieved on CapsGNN which indicates the little effectiveness of Coordinate Addition Module in these two datasets. More specifically, the comparison between CapsGNN-Avg-noRout and CapsGNN-Avg on NCI1, PROTEINS and D&D indicates the effectiveness of Routing mechanism which improves the accuracy by 2.1%, 0.6% and 0.51% respectively. Besides, the comparison between CapsGNN-Avg-noRout and CapsGNN-noRout on NCI1, PROTEINS and D&D indicates the effectiveness of Attention Module which improves the accuracy by 1.37%, 1.24% and 1.35% respectively. The accuracy of NCI1, PROTEINS and D&D can be improved by as much as 2.98%, 1.33% and 2.23% when Attention Module and Routing mechanism are combined in the architecture.

Overall, CapsGNN is a general framework that fuses capsule theory to GNN for more efficient graph representation learning. In this framework, we also provide multiple possible modules to improve the quality of learned graph embeddings while we do not target to find the best combination of modules for each dataset. Since each possible module plays a different role in different datasets, it would be better to adjust the architecture and hyper-parameters based on practical situation.

