# OpenReview forum: "Capsule Graph Neural Network"
_ICLR.cc/2019/Conference_

### Official Review · AnonReviewer2 · 2018-10-31
**A long paper with incomplete experiments**

**Rating:** 6
**Confidence:** 4

**Review:**

The paper fuses Capsule Networks with Graph Neural Networks. The idea seems technically correct and is well-written. With 13 pages the paper seems really long. Moreover, the experimental part seems to be too short. So, the theoretical and experimental part is not well-balanced.

Minor concerns/ notes to the authors:
1.	Page 1: The abbreviation GNN is used before it is defined.
2.	Page 2: I guess there is a mistake in your indices. Capital N == n or?
3.	Page 4: What is \mathbf{I}? I guess you mean the identity matrix.
4.	Page 4: Could you define/describe C_all?
5.	Page 5: Can you describe how you perform the coordinate addition or add a reference?
6.	Page 6: The idea to use reconstruction as regularization method is not new. May you can add a respective reference?
7.	Page 8: The abbreviations in your result tables are confusing. They are not aligned with the text. For example, what is Caps-CNN for a model?

My major concern is about your experimental evaluation. Under a first look the result tables looking great. But that’s due to fact, that you marked the two best values in bold type. To be more precise, the method WL is in the most cases better than your proposed method. This makes me wondering if there is a real improvement by your method. It would be easier to decide if you would present the training/inference times and the number of parameters. By having that, I could relate your results regarding an accuracy-complexity tradeoff.  Moreover, your t-SNE and attention visualizations are not convincing. As you may know, the output of a t-SNE strongly dependents on the chosen hyper-parameters like the perplexity, etc. You not mentioned the setting of these values. Additionally, it is hard to decide if your embeddings are good or not because you are not presenting a baseline or referencing a respective work. You are complaining that this is due to the space restrictions. But you have unlimited capacity in the appendix. So please provide some clarifying plots. Finally, I’m also not convinced that your attention mechanism works as expected. It’s again due to missing baseline results and/or a reference. If it’s not possible to add one of them, you could perform an easy experiment where you freeze your fully-connected layers of the attention module to fixed values (maybe such that it performs just an averaging) and repeat your experiments. In case your attention module works as expected you should observe a real change in terms of accuracy and in your visualizations too.
You could also think about to publish your code or present further results/plots in a separate blog.

Update:
According to the revised version which addresses a lot of my concerns, I vote for marginally above acceptance threshold.

---

### Official Review · AnonReviewer1 · 2018-11-02
**Capsule networks for graphs without convincing motivation and experimental evaluation**

**Rating:** 6
**Confidence:** 4

**Review:**

The authors provide an architecture that applies recent advances in the field of capsule networks in the graph neural network domain. First, hierarchical node level capsules are extracted using GCN layers. Second, after weighting each capsule by the output of a proposed attention module, graph level capsules are computed by performing a global dynamic routing. These graph level capsules are used for training a capsule classifier using a margin loss and a reconstruction loss.

The general architecture seems to be a reasonable application of the capsule principle in the graph domain, following the proof of concept MNIST architecture proposed by Sabour et al.

My main concern is that I have problems grasping the motivation behind using capsules in the given scenario. Besides an unprecise motivation in the introduction, there is no clear reason why the routing mechanism helps with solving the given tasks. Capsule networks capture pose covariances by applying a linear, trainable transformation to pose vectors and computing the agreement of the resulting votes. It is not clear to me how discrete information like graph connectivity can be encoded in a pose vector so that linear transformations are able to match different "connectivity poses".

Is there a more formal argument that explains why capsules should be able to capture more information about the input graph than other GCNNs?

Also, some design choices seem to be quite arbitrary. One example is using the last feature maps of the GCN as positions for coordinate addition. Is there a theoretical/intuitive motivation for this?

Results for the given experiments show improvement on some graphs. However, the authors proposed several concepts: a global pooling method using dynamic routing, an attention mechanism, a novel reconstruction loss, interpreting deep node embeddings as spatial positions. It is not clear to what extent the individual aspects of the method contribute to the gains. The qualitative capsule embedding analysis is interesting. However, this part needs a comparison to standard global graph embeddings to see if there is a significant difference.

In my opinion, the paper needs:
1) a clear experimental evaluation showing that capsules and the dynamic routing lead to improved results (i.e. by providing an ablation study to show which gains result from the attention-based global pooling mechanism, the reconstruction loss, the dynamic routing and from the coordinate addition), or
2) a more precise motivation for the use of dynamic routing to capture correlation between pose vectors in graphs in general (i.e. formal arguments why the method is stronger in capturing statistics or for what types of graphs it provides more discriminative power).

Overall, the paper does not convince me that capsules and dynamic routing provide advantages if used like the authors propose. Therefore, I tend to voting for rejecting the paper as long as points 1) and 2) are not addressed properly.


Minor remarks:

- There are quite a lot of grammatical errors (especially missing articles).

--------------------------
Update:
The authors addressed some of the weak points mentioned above adequately. The experimental evaluation was significantly improved and the results are a nice contribution. However, the theoretical contribution and the poor motivation of capsules in the graph context remain weak points. I have updated my rating accordingly.

---

### Official Review · AnonReviewer3 · 2018-11-03
**The proposed CapsGNN is original and achieves good results on some datasets; Some more discussions may further help.**

**Rating:** 6
**Confidence:** 4

**Review:**

This paper was written with good quality and clarity. Their idea was original and experiment results show the proposed CapsGNN is effective in large graph data analysis, particularly on graphs with macroscopic properties.

Pros:

1) The paper makes a clear and detailed comparison between the proposed CapsGNN and the related models in section 3.2.

2) Use of capsules nets and routing in CapsGNN are close to that in the original CapsNet, with the core characteristics (and potential advantages) of capsules and dynamic routing being perserved in the proposed CapsGNN to handle the targeted problem.

3) The comparison and model analysis are thorough and comprehensive.

Cons or unclear points:

1) Why the paper does not include all biological datasets (6 datasets in total, only 4 used in this papaer) presented in (Verma & Zhang, 2018) in the experiment section. The experiments in Verma & Zhang, (2018) show that the GCAPS-CNN achieved SOTA results on nearly all biological datasets. Does GCAPS-CNN outperformed CapsGNN on biological datasets? It will be nice if there is comparison on more datasets and more analysis is provided between CapsGNN and GCAPS-CNN.

2) Why CapsGNN is not suitable for preserving information of fine structures? Can the authors give more explanation and discussions?

---

### Public Comment · ~Benedek_Rozemberczki1 · 2019-03-25
**Implementation**

I tried to implement the paper in PyTorch: https://github.com/benedekrozemberczki/CapsGNN.

What was the actual number of epoch usually used in the paper?

---

> ### Author Response · Authors · 2019-03-25
> **Number of epoch**
>
> Hi, just as what is mentioned in the paper. The exact number of epochs depends on the validation accuracy.
>
> When I did the experiments, I conducted 10-fold cross validation and for each fold I will run enough and a same number of epochs so that the models are overfitting on almost all the validation folds.  Then I chose the testing accuracy of the model which achieves the highest accuracy on corresponding validation fold as the final reported accuracy.
>
> So here, I can provide you the largest number of epochs I set for each dataset and you can find the exact number of epochs based on your validation data:
>
>            MUTAG: 2000
>         ENZYMES: 3000
>        PROTEINS: 2000
>                  D&D: 300
>                  NCI1: 1500
>            COLLAB: 300
>            IMDB-B: 2000
>           IMDB-M: 2000 (but usually reach highest within 500 epochs)
>   REDDIT-M5K: 150 (can try more epochs)
> REDDIT-M12K: 150 (can try more epochs)

---

### Author Response · Authors · 2019-04-11
**Implementation**

The Tensorflow implementation is available at https://github.com/XinyiZ001/CapsGNN

---

### Meta-Review · Area_Chair1 · 2018-12-14
**The reviewers hope to see further improvements.**

**Confidence:** 4
**Recommendation:** Accept (Poster)

**Metareview:**

AR1 asks for a clear experimental evaluation showing that capsules and dynamic routing help in the GCN setting. After rebuttal, AR1 seems satisfied that routing in CapsGNN might help generate 'more representative graph embeddings from different aspects'. AC strongly encourages the authors to improve the discussion on these 'different aspects' as currently it feels vague. AR2 is initially concerned about experimental evaluations and whether the attention mechanism works as expected, though, he/she is happy with the revised experiments. AR3 would like to see all biological datasets included in experiments. He/she is also concerned about the lack of ability to preserve fine structures by CapsGNN. The authors leave this aspect of their approach for the future work.

On balance, all reviewers felt this paper is a borderline paper. After going through all questions and responses, AC sees that many requests about aspects of the proposed method have not been clarified by the authors. However, reviewers note that the authors provided more evaluations/visualisations etc. The reviewers expressed hope (numerous times) that this initial attempt to introduce capsules into GCN will result in future developments and  improvements. While AC thinks this is an overoptimistic view, AC will give the authors the benefit of doubt and will advocate a weak accept.

The authors are asked to incorporate all modifications requested by the reviewers. Moreover, 'Graph capsule convolutional neural networks' is not a mere ArXiV work. It is an ICML workshop paper. Kindly check all ArXiV references and update with the actual conference venues.